# Using Glyphosate on Guarana Seedlings in the Amazon

**DOI:** 10.3390/molecules28135193

**Published:** 2023-07-04

**Authors:** Bruna Nogueira Leite Konrad, Sara Cruz Pinheiro, Carla Coelho Ferreira, Evandro Konrad Hoffmann, Sônia Maria Figueiredo Albertino

**Affiliations:** 1Programa de Pós-graduação em Agronomia Tropical, Universidade Federal do Amazonas, Manaus 69067-005, AM, Brazil; brunanleite@hotmail.com (B.N.L.K.); carlacoelhofigueiredo@gmail.com (C.C.F.); 2Faculdade de Ciências Agrárias, Universidade Federal do Amazonas, Manaus 69067-005, AM, Brazil; pinheiroscpcruz@gmail.com; 3Programa de Pós Graduação Agricultura no Trópico Úmido, Instituto Nacional de Pesquisa da Amazônia, Manaus 69060-001, AM, Brazil; konrad.eh@gmail.com

**Keywords:** *Paullinia cupana*, herbicide, Injury, anthracnose

## Abstract

The seed yield of guarana (*Paullinia cupana* H.B.K. var. sorbilis) is affected by weeds. Management is difficult for Amazon farmers and ranchers, owing to the hot and humid climate prevailing in the region, which makes mechanical control inefficient and leads farmers to the decision to use herbicides. Herbicide damage to this species is unknown. The objective of this study was to evaluate glyphosate damage to the development and quality of guarana seedlings. The treatments consisted of glyphosate doses at concentrations of 0, 126, 252, 540, 1080, 2160 and 3240 g a.e. ha^−1^ and were evaluated for 60 days, in two applications. Analyses were performed for biometrics, seedling development, anthracnose and Injury characteristics. Glyphosate caused symptoms of Injury in all doses applied, but lower doses did not interfere with seedling growth and development. There was a correlation between anthracnose severity and increased glyphosate dose. When applied correctly, glyphosate can be an integrated weed management tool for use in guarana crops.

## 1. Introduction

Guarana (*Paullinia cupana* H.B.K. var. sorbilis) is known for having seeds with high levels of caffeine and tannins, which gives this species highly characteristic pharmacological and energy features [1]. Brazil is the only country that commercially produces guarana, whose seeds are widely known for their benefits and are used in a wide range of industries, from beverages to pharmaceuticals [2]. For years, the crop has been domesticated, and its management has been modernized to increase production and reduce the impact caused by diseases, pests and weeds [3]. Weed control, in particular, has many gaps to be filled to improve crop management.

Weed interference reduces guarana production by up to 65%. Therefore, weed management is necessary to enable production [4]. The use of herbicides has advantages over mechanical control since it is more efficient in controlling aggressive weed species and in high air relative humidity and abundant rainfall conditions. However, the inappropriate use of these products may affect the physiology and development of non-target organisms [5]. Even when applied to target plants, herbicides may be blown by the wind and hit the crop by drift [6]. This stress caused by drift has been widely studied. Glyphosate is the main herbicide used worldwide, especially in grain crops; excessive use of it may cause physiological stress in non-target plants [7]. Because it acts on the shikimate pathway, it affects the synthesis of important aromatic amino acids that particularly act on plant defense characteristics [8]. Previous studies have shown that low doses of glyphosate affected the development of vegetables such as lettuce, tomatoes, pumpkin, pepper, sunflower and non-transgenic soybeans [9] as well as of cultivated forest species such as eucalyptus [10].

The stress caused by these low doses also increases susceptibility to diseases, either by affecting the plant defenses or by acting on phytopathogenic microorganisms, because some microorganisms have the enzyme EPSPs [11]. The presence of glyphosate inside the plants also increases the amount of ammonia, glutamine and glutamate to phytotoxic levels, which can lead to plant death. For this reason, the main features of Injury caused by glyphosate are total or partial yellowing of plants [12].

Intensive weed management in guarana plants occurs in the first two years after planting to stop competition from hindering the establishment of the crop in the field. The use of glyphosate is recommended in cases of high infestation and in the presence of aggressive weed species [13]. It is considered as the most efficient method in terms of control and cost, especially in large areas, where mechanical control becomes infeasible [14]. 

However, the possible effect of glyphosate on guarana plants if it is spread by drift is still not known. Thus, the objective of this study was to evaluate the effects of different doses of glyphosate on the biometric characteristics of guarana seedlings, injury and anthracnose severity.

## 2. Results

### 2.1. Biometric Measurements

ANOVA indicated the interaction between the dose and time variables for all biometric characteristics in the first application of the herbicide; therefore, regression analysis was performed for characteristics with a significant interaction. However, there was no interaction between the dose and time variables for any of the characteristics analyzed in the second application (Table 1).

The doses of 2160 g and 3240 g acid equivalent (a.e.) ha^−1^ reduced the number of leaves (NL) and leaf area (LA) of the guarana seedlings. At 60 days after application (DAA), the dose of 2160 g a.e. ha^−1^ reduced NL by 57% and LA by 63%, while the dose of 3240 g a.e. ha^−1^ ha^−1^ reduced NL and LA by 100% (Figure 1A,B). Only the dose of 3240 g a.e. ha^−1^ affected seedling height and stem diameter of guarana (Figure 1C,D). The doses of 2160 g a.e. ha^−1^ and 3240 g a.e. ha^−1^ correspond to 100 and 150% of the recommended glyphosate dose for desiccation and mowing, respectively. The limit supported by the seedlings was 1080 g a.e. ha^−1^, which was confirmed in the second application of glyphosate, in which the plants remained undamaged (Figure 2).

### 2.2. Plant Injury (Phytotoxicity)

ANOVA indicated the interaction between the dose and time variable for Injury in the first and second applications of the herbicide; therefore, regression analysis was performed (Table 2). Pearson’s correlation analysis was performed between time and dose factors to analyze Injury in both applications. There was a strong correlation between the factors in the first application (correlation = 0.88 *) and a moderate correlation in the second application (correlation = 0.69 *).

#### 2.2.1. First Application

At 14 DAA, death re-sprouting tissue occurred in the seedlings that had received the dose of 3240 g a.e. ha^−1^, followed by yellowing of young leaves and necrosis in the veins (Figure 3A,B). At the end of the evaluations, the seedlings from this treatment lost all leaves and ultimately died.

For the seedlings under the dose of 2160 g a.e. ha^−1^, yellowing of young leaves occurred at 14 DAA. At 30 DAA, there was a loss of young leaves and tips and an occurrence of stains on the older leaves. At 60 DAA, there was a loss of almost 50% of the seedlings (Figure 3C).

The doses of 1080 g a.e. ha^−1^ and 540 g a.e. ha^−1^ also caused yellowing of young leaves at 14 DAA. However, there was no significant loss of leaves and tips. At 60 DAA, there were yellow stains on the edges of the older leaves (Figure 3D).

Injury to guarana seedlings was proportional to the increase in glyphosate dose and were visible as of 7 DAA. There were high and very high levels of injury only at the doses of 2160 g a.e. ha^−1^ and 3240 g a.e. ha^−1^ (Figure 3A,D). In both cases, injuries were irreversible. The dose of 3240 g caused the death of 100% of the plants at 60 DAA, and the dose of 2160 g caused 50% death and severe damage to the seedlings at the end of the evaluations (Figure 4).

The doses of 126 g and 252 g a.e. ha^−1^ caused mild, low-intensity injuries (Figure 3E,F). The dose of 540 g a.e. ha^−1^ caused acceptable symptoms that were tolerated by the plants. However, the dose of 1080 g a.e. ha^−1^ presented worrisome Injury, according to the grading scale proposed by the Brazilian Society of Weed Science for plants under herbicide doses. The worrying grade on the scale is a more drastic, but still recoverable, symptomatology (Figure 4).

#### 2.2.2. Second Application

The doses of 540 g and 1080 g a.e. ha^−1^ caused narrowing in both the young and the fully expanded leaves, in addition to changes in the shape of the leaf borders and depigmentation along the leaflets (Figure 5A,B). After the doses of 126 g and 252 g a.e. ha^−1^ were applied, guarana seedlings presented depigmentation of older leaves, abnormal color, wavy leaf blade edges and wrinkled leaves, but new, healthy leaves were produced at 60 DAA (Figure 5C,D). The symptoms were worrying, but the plants recovered from them according to the Injury analysis (Figure 6).

### 2.3. Anthracnose Severity

Analysis of variance indicated an interaction between dose and time factors in the first application of glyphosate in guarana seedlings. There was no significant interaction in the second application, despite the occurrence of anthracnose in the treated seedlings (Table 3).

The occurrence of anthracnose started 14 days after the first application. The highest level of severity of the disease was found in seedlings treated with the highest doses of glyphosate. The seedlings from the control treatment and those under a dose of 262 g a.e. ha^−1^ did not present anthracnose (Figure 7). The doses of 2160 and 3240 g a.e. ha^−1^ exceeded the severity of anthracnose in the seedlings by more than 50% (Figure 8).

## 3. Discussion

Understanding the action of different doses of glyphosate on growth, phytotoxic and sanitary characteristics of guarana seedlings can help to prevent undesirable effects of the herbicide on crop management and increase knowledge about the cultivation of the species. 

Low doses of glyphosate (≤1080 g a.e. ha^−1^) did not irreversibly affect the development of guarana seedlings (Figure 2), but they impacted leaf morphology and seedling health (Figure 3, Figure 4, Figure 5, Figure 6, Figure 7 and Figure 8). Glyphosate acts on the shikimate pathway, damaging the synthesis of the aromatic amino acids phenylalanine and tyrosine, which are related to plant defense, and tryptophan, a precursor of the hormone AIA, which, in turn, is linked to plant growth [15]. Thus, studies on the effects of glyphosate on plants should approach aspects of plant growth and plant health.

Importantly, the higher doses applied to the guarana seedlings in this experiment are higher than those in other studies with perennial species. In an experiment with coffee plants, the highest dose applied was 460.8 g a.e. ha^−1^, and it affected leaf area owing to leaf narrowing as a response to glyphosate Injury [16]. Similarly, in rubber tree seedlings, the dose of 345.6 g a.e. ha^−1^ was enough to reduce the height and diameter of seedling stems [17]. In the present study, the low doses did not change the number of leaves, shoot branching or stem diameter.

The presence or absence of damage caused by low doses of glyphosate to the initial growth of the species, may be due to different genotypes or clones and their metabolic variations and gas exchange. In an experiment with eucalyptus plantations, the reduction in stem height and stem diameter for the *Eucalyptus grandis* clone in comparison to *Eucalyptus urophylla* was partly due to a reduction in gas exchange [18]. This may explain why low doses of glyphosate in this experiment did not hinder the initial development of guarana seedlings, as an experiment with two guarana cultivars showed that low doses of glyphosate caused an increase in gas exchange for the BRS-Maués cultivar, but a reduction for BRS-Andirá [19].

Thus, the effect of low doses of glyphosate depends on many intrinsic factors of the species and may not cause damage to the initial development of non-domesticated forest species, as found in an experiment with *Acacia polyphylla*, *Enterolobium contortisiliquum*, *Ceiba speciosa* and *Luehea divaricata* with glyphosate doses lower than 2160 g a.e. ha^−1^ [20]. Guarana is a semi-domesticated species whose secondary metabolism has particular characteristics that may increase stress tolerance. For example, the leaves of guarana plants have high concentrations of methylxanthines, which are alkaloid substances related to plant protection [21]. This may explain the detoxification capacity of the seedlings that were negatively affected by the action of glyphosate.

The thick cuticular layer of the leaves of guarana plants [22] can also influence the translocation of glyphosate as it acts as a protective barrier against external agents, reducing the absorption of glyphosate [23]. Also, the dose at low concentrations results in a lower amount of translocated product in the plant [24], which reduces herbicidal activity. It can be assumed that these characteristics, which are known to be present in guarana plants, combined with low doses of glyphosate may lead to lower translocation of glyphosate in the plant; consequently, damage may not occur to the initial development of the species.

Another characteristic that affects the effect of glyphosate on plants is the growth stage. In studies carried out with *Brassica napus*, it was found that after herbicide application, the seedlings directed the energy produced during photosynthesis to mechanisms to protect the photosynthetic apparatus, thereby preserving the accumulation of energy, which is essential for plant development [25]. As the studies were carried out in young guarana plants, the same may have occurred.

The symptoms of Injury found in the guarana seedlings in the present study were expected for plants affected by herbicidal activity. The progress of the symptoms occurs initially with yellowing of the younger leaves, and even of the older ones, followed by wilt, necrosis and consequently plant death within 4 to 20 days, depending on species and plant health. Other common symptoms are wrinkling, leaf malformations and excess sprouting in regrowth areas of the plants [26]. In studies with coffee seedlings, the authors related chlorosis to chlorophyll reduction, and leaf blade hardening to cell dysfunction caused by the herbicide [27]. However, chlorosis may be related to the accumulation of phytotoxic levels of ammonia, glutamine and glutamate [12].

Leaf malformations and excess leaf sprouts may also be related to the deregulation of growth hormones [28]. However, even if the plants show symptoms of Injury, their development is not compromised, as found in a study with peach seedlings: the symptoms of Injury were proportional to the increase in doses (0.178 g, 356 g, 712 g and 1424 g a.e. ha^−1^), but did not impair the growth characteristics of the seedlings [29]. Even though the low doses of glyphosate showed symptoms of Injury, the development of the seedlings did not change.

Another important aspect of the action of glyphosate is plant defense mechanisms, which result in the susceptibility of species to diseases. Importantly, guarana seedlings, in this experiment, did not have anthracnose symptoms or any other disease before the application of the herbicide glyphosate; still, anthracnose occurred and was proportional to the increase in the doses of glyphosate (Figure 7).

The fungi of the genus *Colletotrichum* are found with different life habits and are usually considered phytopathogens; however, the same species may act as a pathogen or an endophyte [30]. Thus, even if the guarana seedlings had not been injured before the application of glyphosate, the action of the herbicide may have favored the virulence of the agent. In unpublished experiments, it was found that low doses of glyphosate increased the growth of these fungi in a controlled environment.

The guarana cultivar BRS-Maués was developed by the Genetic Breeding Program for Guarana at EMBRAPA Western Amazon. It is considered as tolerant to anthracnose; thus, it is assumed that glyphosate broke tolerance to anthracnose in the plants of the present study.

The residual action of glyphosate affects the interactions between plants and heterotrophic organisms, substantially interfering with plant resistance. In many crops, the constant application of glyphosate is related to an increase in disease severity [31]. In addition, it may also cause immobilization of specific micronutrients involved in disease resistance, and a reduction in plant growth and vigor [32]. Even herbicide-resistant crops may be susceptible to diseases. The herbicide may interact with fungi and alter their mechanisms; some microorganisms have the shikimate pathway and, therefore, may be influenced by the positive or negative effects of the herbicide [33].

All phytotoxic effects of glyphosate are produced by the inhibition of the 5-enolpyruvylshikimate-3-phosphate synthase (EPSPS) enzyme. Usually, its effects on non-target plants are smaller in comparison to those of many other herbicides, owing to its composition, application methods and strong binding to most soils, which ultimately inactivate it. Still, there may be varied actions on pathogens, e.g., promoting disease control or even changing virulence [34].

Although the low doses of glyphosate have not affected the development of the guarana seedlings, it is worth noting that in woody plant species, glyphosate may persist in plant tissues for up to a decade, even if applied in doses considered as sublethal [35], and these residues can be translocated to the fruits.

## 4. Material and Methods

### 4.1. Plant Material and Growth Conditions

One-year-old seedlings of the guarana plant (*Paullinia cupana* H.B.K. var. Sorbilis (Mart.), cultivar BRS-Maués Ducke) were donated by the company Agropecuária Jayoro (Presidente Figueiredo, Amazonas, Brazil) and were kept in a shaded nursery at 50% for three months. After this period, they were transferred to a greenhouse, with controlled temperature and water, but without shading. They remained in this environment for nine months for acclimatization; they were transplanted twice to ensure uniform size, compliance with sanitary requirements and adequate nutritional status.

Fertilization before the experimental setup was based on chemical analyses of the soil and the leaves of the seedlings (Table 4). The recommendation followed basic fertilization guidelines for general crops in general [36] and for forest species [37] since there is no specific recommendation for the cultivation of guarana plants in the juvenile stage. First, 2.13 kg of dolomitic lime was added to the substrate per m^3^, and after fifteen days of incubation, the fertilizers were incorporated. The sources and concentrations used were as follows: urea (250 g N. m^−3^), triple superphosphate (900 g P_2_O_5_. m^−3^), potassium chloride (180 g K_2_O. m^−3^) and FTEBR12 (200 g micronutrients. m^−3^).

The conditions of the controlled environment throughout the experiment were relative humidity = 74 RH, temperature = 31 °C and irradiance = 565 μmol m^−2^ s^−1^. Irrigation was performed by means of automatic sprinklers, twice a day with a duration of 10 min or according to the daily requirement.

### 4.2. Herbicide and Application Information

The experiment used the herbicide PESTANAL, analytic standard (99% purity), glyphosate, (Sigma-Aldrich^®^, Brazil) and the concentrations were based on the maximum value recommended for area desiccation, which is 6 L of the commercial product (p.c) per hectare (ha), considering aggressive weed infestations in the field. The acid equivalent (a.e.) base was the concentration of 360 g/L, which is found most frequently on the market.

The application of the doses followed the established premises for spraying; it was performed with a Herbicat electric backpack sprayer (Turia Duo 18 L), with a maximum flow of 1.8 L/min, using a fan nozzle (110:02,TeeJet^®^ Technologies, Brazil), at 2 bar pressure and spray volume of 200 L ha^−1^. The application of the herbicide was performed at 20 cm above the top of the seedlings.

### 4.3. Experimental Design

The experimental design was completely randomized. In the first application, seven treatments (doses) were evaluated in six periods (7 × 6), with four replications. The following doses were used: 0, 126, 252, 540, 1080, 2160 and 3240 g of acid equivalent (a.e. ha^−1^) plus sticker-spreaders (AGRAL^®^, Syngenta, Brazil) at a concentration of 0.5% in relation to the spray. The response variables were recorded at 0, 3, 7, 14, 30 and 60 days after the application of the herbicide (DAA). The second application of the herbicide was performed only with the plants that survived the treatments of the first application of glyphosate. The doses were 0, 126, 252, 540 and 1080 g a.e. ha^−1^ plus AGRAL at 7, 14, 30 and 60 DAA, in a 5 × 4 arrangement, with four repetitions. There were 8 specimens for each treatment (dose): two plants per experimental unit.

### 4.4. Data Collection

In each evaluation period, the following growth responses were recorded: height (from base to first leaf insertion); stem diameter (cm), measured with a digital pachymeter; leaf area (cm^2^), measured with a portable meter (L I-3000C, LI-COR Biosciences, USA). Injury was visually analyzed using a scale of grades developed for plants receiving doses of the herbicide glyphosate [38], where 0 means no symptoms and 100 means plant death. Anthracnose severity was evaluated using a diagrammatic scale of injury percentages [39]. Koch’s postulates were applied to confirm the disease. Later, guarana seedlings were inoculated for validation of the pathogenicity of the fungus.

### 4.5. Data Analysis

All data underwent the Shapiro–Wilk test for normality. Non-normal data underwent square root transformation. Variance analysis was performed at 0.05 probability. The data with statistically significant differences underwent regression analysis, using the equation that best fit the original data combined with the biological explanation of the characteristic. The free software package ExpDes.pt R (version 4.0.2) was used.

For Injury data, Pearson’s correlation test between dose and time factors was performed using the free software package Dplyr.pt (version 4.0.2).

Anthracnose incidence data did not undergo regression, but they were expressed in a disease progress curve, designed using the average percentage of anthracnose incidence versus time. The Lattice.pt and LatticeExtra.pt packages of free software R (version 4.0.2) were used.

## 5. Conclusions

Glyphosate caused visual symptoms of Injury in the leaves of guarana seedlings; however, at low doses, it did not interfere with plant growth for the BRS-Maués cultivar.

The action of glyphosate increased the incidence of anthracnose in guarana seedlings of the BRS-Maués cultivar.

When correctly applied, glyphosate can be used as a tool for integrated weed management in guarana crops without causing damage to their development. However, more studies are needed to analyze the persistence of glyphosate and possible residues in guarana seeds.

## Figures and Tables

**Figure 1 molecules-28-05193-f001:**
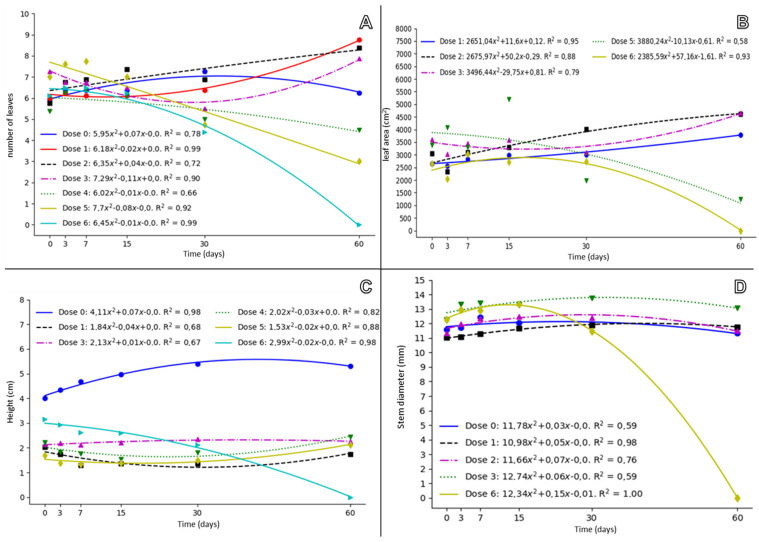
Number of leaves (**A**), leaf area (**B**), height (**C**) and stem diameter (**D**) of guarana seedlings, cultivar BRS-Maués, under different doses of glyphosate, evaluated at 60 days after application. Manaus-Am, 2019. Dose 1 (0 g), Dose 2 (126 g), Dose 3 (252 g), Dose 4 (540 g), Dose 5 (1080 g), Dose 6 (2160 g) and Dose 7 (3240 g) ha^−1^.

**Figure 2 molecules-28-05193-f002:**
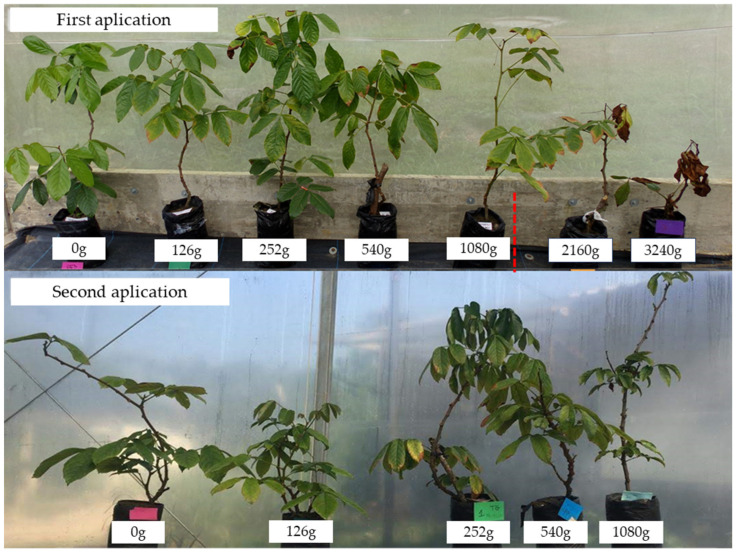
Guarana seedlings, cultivar BRS-Maués, under different doses (g a.e. ha^−1^) of glyphosate at 60 days after application. Manaus-Am, 2019.

**Figure 3 molecules-28-05193-f003:**
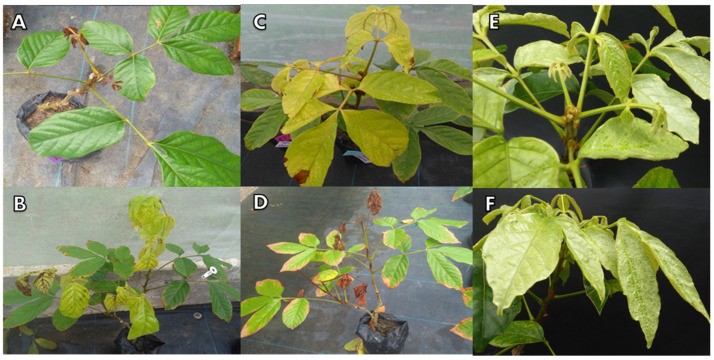
Symptoms of injury to guarana seedlings, cultivar BRS-Maués, after glyphosate application. Manaus-Am, 2019. (**A**) Death re-sprouting tissue; (**B**) Necrosis of between the veins of the leaves; (**C**) New leaves with yellow discoloration; (**D**) Necrosis on the outer edges of the leaves; (**E**) White discoloration of the leaves and irregular growth point; and (**F**) Curling of leaf tissues.

**Figure 4 molecules-28-05193-f004:**
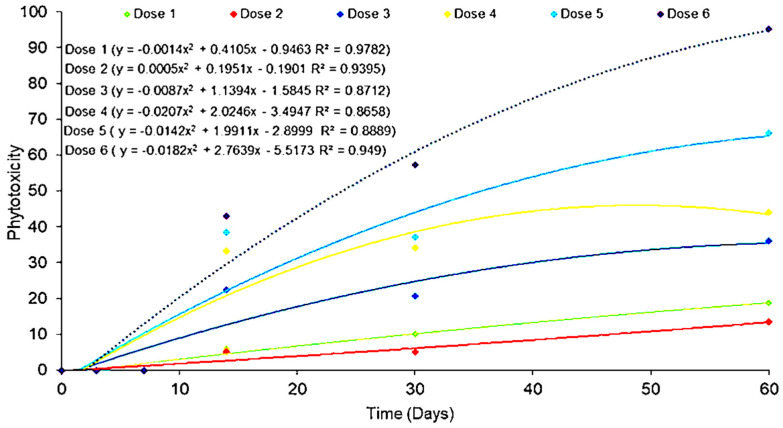
Phytotoxicity in guarana seedlings, cultivar BRS-Maués, after glyphosate application, evaluated for 60 days. Manaus-Am, 2019. Dose 1 (0 g), Dose 2 (126 g), Dose 3 (252 g), Dose 4 (540 g), Dose 5 (1080 g), Dose 6 (2160 g) and Dose 7 (3240 g) ha^−1^.

**Figure 5 molecules-28-05193-f005:**
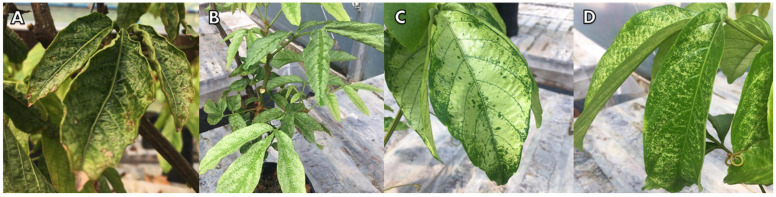
Injury characteristics in guarana seedlings, BRS-Maués cultivar, at 60 days after the application of glyphosate doses. Manaus-Am, 2019. (**A**) Curling of leaf tissues; (**B**) Re-sprouting tissue chlorotic and malformed; (**C**) White discoloration of the leaves; and (**D**) White spots in a spray pattern.

**Figure 6 molecules-28-05193-f006:**
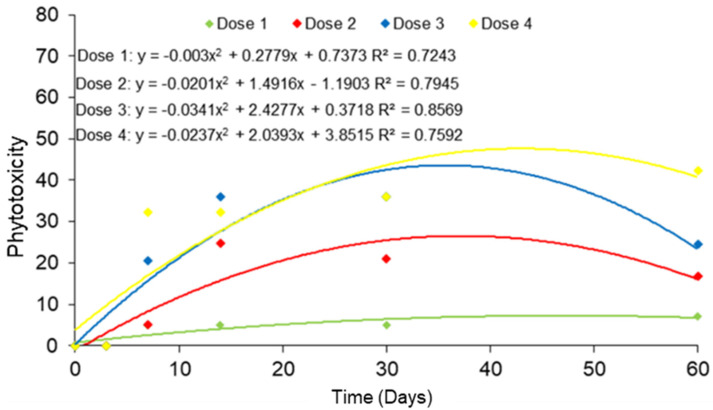
Phytotoxicity in guarana seedlings, BRS-Maués cultivar, after application of glyphosate doses, evaluated at 60 DAA. Manaus-Am, 2019. Dose 1 (0 g), Dose 2 (126 g), Dose 3 (252 g) and Dose 4 (540 g) a.e. ha^−1^.

**Figure 7 molecules-28-05193-f007:**
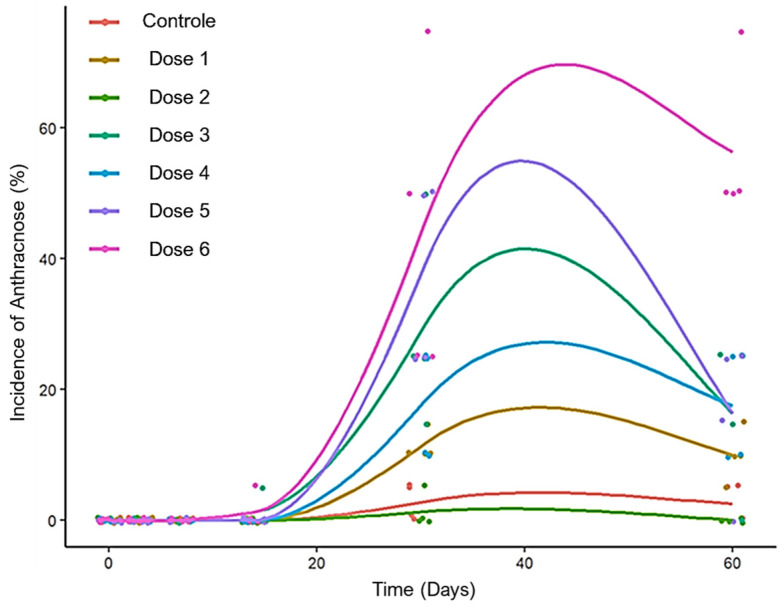
Anthracnose progress curve, determined on the basis of disease severity in BRS-Maués cultivar seedlings that had received doses of glyphosate and were observed for 60 days after application. Manaus-Am, 2019. Control (0 g), Dose 1 (126 g), Dose 2 (252 g), Dose 3 (540 g) Dose 4 (1080 g), Dose 5 (2160 g) and Dose 6 (3240 g) a.e. ha^−1^.

**Figure 8 molecules-28-05193-f008:**
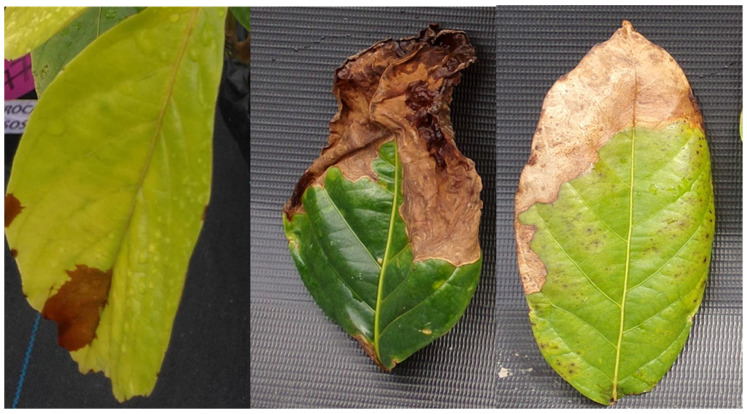
Stains caused by the incidence of *Colletotrichum* spp. in new leaves of guarana seedlings, cultivar BRS-Maués, after application of glyphosate doses. Manaus-Am, 2019.

**Table 1 molecules-28-05193-t001:** Summary of ANOVA for number of leaves (NL), seedling height (H), stem diameter (D) and leaf area (LA) of guarana seedlings (BRS-Maués cultivar), under different doses of glyphosate, analyzed for 60 days after application (DAA). Manaus-Am, 2020.

VF	DF	MS			
		NL	H (cm)	D (cm^3^)	LA (cm^2^)
Application 1					
Time	5	14.34 *	10.60 *	58.59 *	555.88 *
Dose	6	15.07 *	12.81 *	39.12 *	1193.52 *
Time × Dose	30	13.56 *	11.05 *	44.13 *	523.13 *
Residues	126	0.1425	0.15	2.24	114.15
CV%		13.94	22.51	12.91	19.43
Application 2					
Time	3	68.03 *	3.42 ns	29.81 *	3025.49 ns
Dose	4	42.92 ns	30.50 *	6.90 ns	8294.04 *
Time × Dose	12	18.83 ns	2.07 ns	3.52 ns	2232.53 ns
Residues	90	19.99	4.49	4.29	3329.07
CV%		44.63	50.72	15.45	36.91

* = significant at the 0.05 probability level; ns = non-significant.

**Table 2 molecules-28-05193-t002:** Summary of ANOVA for Injury of guarana seedlings (BRS-Maués cultivar), under different doses of glyphosate, analyzed for 60 days after application (DAA). Manaus-Am, 2020.

FV	DF	MS
Application 1		
Time	3	3 *
Doses	6	2 *
Time × Doses	18	4 *
Residues	196	5
CV%	-	29.72
Application 2		
Time	3	2 *
Doses	4	4 *
Time × Doses	12	3 *
Residues	60	5
CV%	-	15.2

* = significant at the 0.05 probability level.

**Table 3 molecules-28-05193-t003:** Summary of ANOVA results for anthracnose severity in guarana seedlings (BRS-Maués cultivar) after application of glyphosate doses, evaluated at 60 DAA. Manaus-Am, 2020.

VF	DF	MS
Application 1		
Time	1	9119.00 ***
Doses	6	781.00 ***
Time × Doses	6	1309.40 ***
Residues	154	74,200
Adjusted R2	-	0.62
Application 2		
Time	1	526.27 *
Doses	1	54.95 ns
Time × Doses	1	93.63 ns
Residues	236	79.11
Adjusted R2	-	0.02

*** = significant at the 0.01 probability level; * = significant at the 0.05 probability level; ns = non-significant.

**Table 4 molecules-28-05193-t004:** Result of chemical and physical analyses of the compost and the leaves of the guarana seedlings. Manaus-Am, 2019.

Substrate										
PH	m	V	H + Al	t	T	SB	Ca	Al	Mg	K
H_2_O	%		cmolc dm^−3^							
4.4	35	12	3	1.63	8.86	1.06	2.6	0.57	0.33	0.08
Total sand	Silt	Clay	O.M.	P	B	Cu	Fe	Mn	Zn	S
g Kg^−1^			dg Kg^−1^	mg dm^−3^						
325	25	650	3.3	6.3	0.21	1.4	75	1.2	3.2	45
Leaves										
N	P	K	Ca	Mg	S	Mn	Zn	B	Cu	Fe
g kg^−1^					mg kg^−1^					
17.3	3.1	10.9	5.1	1.6	1.5	41.1	31.4	68.1	5.9	172.3

## Data Availability

The data used for this research are presented in Table 1, Table 2 and Table 3.

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
