# Peer review of "Using Glyphosate on Guarana Seedlings in the Amazon"

_molecules, 2023, doi:10.3390/molecules28135193_

Round 1
Reviewer 1 Report
This paper deals with the impacts of glyphosate on guarana seedlings in the Amazon.
This reviewer have thoroughly review the subject matter of the manuscript. In my judgement the submission lacks of interest and robustness. The initial hypothesis is to use a herbicide that does not have authorization to be used in guarana. Starting at this point, the study is not useful nor compelling.
The main inconvenience of the manuscript is why the authors performed experiments if glyphosate is not register for guarana crop. From now on, many questions arise that have not been tackled. For example, the dose chosen belongs to area desiccation while the objective is for other use. Thus, the starting point seems wrong.
Thus, the authors tried to establish a pest management without taking into account the GAP (application doses, number of applications..). so preliminary studies establishing a GAP should be needed before performing a risk management.
The objectives and initial hypothesis is not robust and enough justified from a scientific point of view, so the risk management is not useful.
Furthermore, the methodology lacks robustness. It is not clear why the authors use the analytical grade standard with a spreader and do not employ the formulated product.
Which are the weeds that the authors pretend control? Which are the effective dose for these weeds? Which is the phenological stage of the crop to apply the herbicide to control the weeds? Is a post- or pre-emergence use? What are the criteria for the doses employed? All these issues are very important and decisive in the design and none of them has been addressed.
Furthermore, the results are only based on visual criteria and for this reviewer the results obtained are not enough precise and lack of accuracy and robustness.
In summary, this reviewer thinks that this is not an enough interesting and compelling work with objectives relevant to the field.
From this reviewer´s point of view, this work fits better in a agronomic informative journal, but not one of the high quality and standards such as Molecules.
In conclusion, this reviewer does not recommend the publication of the manuscript in this Journal.
Author Response
Dear reviewer,
here is our response to your comments:
Ponto 1:
This paper deals with the impacts of glyphosate on guarana seedlings in the Amazon.
This reviewer have thoroughly review the subject matter of the manuscript. In my judgement the submission lacks of interest and robustness. The initial hypothesis is to use a herbicide that does not have authorization to be used in guarana. Starting at this point, the study is not useful nor compelling.
The main inconvenience of the manuscript is why the authors performed experiments if glyphosate is not register for guarana crop. From now on, many questions arise that have not been tackled. For example, the dose chosen belongs to area desiccation while the objective is for other use. Thus, the starting point seems wrong.
Response 1: The guarana product is known worldwide for its unique energy principles, however, its management plan is still rudimentary and many gaps need to be filled. The study presented in this manuscript seems simple, but its results have an impact on the cultivation of the species and alert for studies such as, for example, on residues that may be contained in the final product of the guarana tree and how much this can impact its commercialization.
They also provide subsidies for adjusting the management of weeds and warn about the inappropriate use of glyphosate and other herbicides which, despite not being registered, are recommended for semi-domestic crops, and are widely used by farmers in the cultivation of guarana in the Amazon, often indiscriminately. In this context, this study corroborates the regulation of glyphosate use in guarana. We would like to clarify that, in the second quarter of 2023, some regulatory changes allowed the use of glyphosate for this crop and recommended doses, considering the acid equivalent of the only product released, which are compatible with those used in this study.
As for the robustness of the document, we made significant adjustments and we hope to meet expectations and clarify the intentions of the study.
Point 2: Thus, the authors tried to establish a pest management without taking into account the GAP (application doses, number of applications..). so preliminary studies establishing a GAP should be needed before performing a risk management.
Response 2: In the cultivation of guarana plant, there is a recommendation for at least four weed controls per year, in the first two years of implementation of the crop in the field. Because the competition of weeds in this period compromises the plantation establishment. Those information can be found in the technical circulars of the institutions responsible for developing crop management, but it is not scientifically published.
As researchers in the area of weed control and knowing the reality that are not exposed scientifically, mitigating the risks of frequent and habitual use of glyphosate seems urgent to us, including assisting in a management proposal.
Point 3: The objectives and initial hypothesis is not robust and enough justified from a scientific point of view, so the risk management is not useful. Furthermore, the methodology lacks robustness. It is not clear why the authors use the analytical grade standard with a spreader and do not employ the formulated product. Which are the weeds that the authors pretend control? Which are the effective dose for these weeds? Which is the phenological stage of the crop to apply the herbicide to control the weeds? Is a post- or pre-emergence use? What are the criteria for the doses employed? All these issues are very important and decisive in the design and none of them has been addressed.
Response 3: We made adjustments to the text of the manuscript that can explain the choice of evaluations for data collection. The management of the crop has many demands and studies thus contribute to the decision making in the control of weeds and has great relevance as a warning to farmers.
As for the product, we chose the analytical grade and the spreader, as there were no commercial products registered at the time of the implementation of the experiment, and by choosing glyphosate in this formulation, we are sure that the results are in fact from the action of glyphosate.
Point 4: Furthermore, the results are only based on visual criteria and for this reviewer the results obtained are not enough precise and lack of accuracy and robustness. In summary, this reviewer thinks that this is not an enough interesting and compelling work with objectives relevant to the field. From this reviewer´s point of view, this work fits better in a agronomic informative journal, but not one of the high quality and standards such as Molecules. In conclusion, this reviewer does not recommend the publication of the manuscript in this Journal.
Response 4:
The main signs of phytotoxicity of glyphosate are those that visually assess and interfere with leaf area and crop growth. Without symptoms or release to Guarana, our study is the first you said I used to investigate the development premise of the study, affecting the production of melons, and how this affects the marketing of the product.
We appreciate your evaluation of the manuscript and hope that with the adjustments made, we can reach the level of the journal.
Reviewer 2 Report
Amendments advised:
Lines 5, 6, 7, 12: Remove hyphens
Line 23: Insert spacing between words
Line 46: Replace the sentence with: "However, the risk posed by glyphosate drift onto non-target species is an unknown factor."
Line 49: Insert "for" between "systems" and "promoting"
Line 66: In the Abstract there is space between values and unit "g", but not here -- adhere strictly to style guideline
Line 71: "2.160" must be "2,160"
Line 73: Spacing between sentences required
Line 74: Replace "supported" with either "withstood" or "tolerated"
Line 79: Replace "submitted" with "exposed" -- do the same throughout the text
Line: 91: Insert "that" between "seedlings" and "received"
Line 97: Insert "yellowish" in front of "spots" ?
Line 145: Inconsistency in style -- here there is space between value and unit "g". Elsewhere in the text there is no space.
Line 166: Inconsistency in style
Line 170: Replace "observed" with "tested" ?
Line 181: Add at end of sentence: "and the local environment" after ".....to the species"
Line 245 to 257: Discussion here is one-sided (apparently biased). Authors are strongly advised to convey a more balanced discussion by referring to the book chapter by S O Duke:
“Glyphosate: Uses Other Than in Glyphosate-Resistant Crops, Mode of Action, Degradation in Plants, and Effects on Non-target Plants and Agricultural Microbes”. Stephen O. Duke (2020).
In: J. B. Knaak (ed.), Reviews of Environmental Contamination and Toxicology Volume 255, Reviews of Environmental Contamination and Toxicology 255, https://doi.org/10.1007/398_2020_53
Line 262: Remove "the" in front of "glyphosate" Line 263: Insert "thereby promoting" in place of "providing" Line 268: Spacing Line 272: Replace "submitted" with "exposed" -- do same elsewhere Line 273: Replace "It" with "Watering was done by means of two ....." Line 276: Spacing Line 277: "We used glyphosate ......." Line 278/9: Spacing between values and units -- inconsistent style! Line 332: Remove "The" in front of "glyphosate" Lines >345: Reference list -- there are many spacing and style issues to correct SUGGESTION for future research on guarana: Authors are made aware of the real possibility that glyphosate residue may accumulate in the guarana product as a result of its use for weed control in the crop. Glyphosate residue in the product could result in rejection of product in the EU, for example, where strict MRL (maximum residue level) regulations exist -- REFER ABOVE BOOK CHAPTER BY S.O. DUKE
Minor amendment advised -- also see the referee's editing
Author Response
Dear reviewer,
here is our response to your comments:
Point 1:
Lines 5, 6, 7, 12: Remove hyphens
Line 23: Insert spacing between words
Line 46: Replace the sentence with: "However, the risk posed by glyphosate drift onto non-target species is an unknown factor."
Line 49: Insert "for" between "systems" and "promoting"
Line 66: In the Abstract there is space between values and unit "g", but not here -- adhere strictly to style guideline
Line 71: "2.160" must be "2,160"
Line 73: Spacing between sentences required
Line 74: Replace "supported" with either "withstood" or "tolerated"
Line 79: Replace "submitted" with "exposed" -- do the same throughout the text
Line: 91: Insert "that" between "seedlings" and "received"
Line 97: Insert "yellowish" in front of "spots" ?
Line 145: Inconsistency in style -- here there is space between value and unit "g". Elsewhere in the text there is no space.
Line 166: Inconsistency in style
Line 170: Replace "observed" with "tested" ?
Line 181: Add at end of sentence: "and the local environment" after ".....to the species"
Line 262: Remove "the" in front of "glyphosate" Line 263: Insert "thereby promoting" in place of "providing" Line 268: Spacing Line 272: Replace "submitted" with "exposed" -- do same elsewhere Line 273: Replace "It" with "Watering was done by means of two ....." Line 276: Spacing Line 277: "We used glyphosate ......." Line 278/9: Spacing between values and units -- inconsistent style! Line 332: Remove "The" in front of "glyphosate" Lines >345: Reference list -- there are many spacing and style issues to correct
Response 1: We've considered all the fine-tuning pointed out in the review and hope we've lived up to your expectations.
Point 2: Line 245 to 257: Discussion here is one-sided (apparently biased). Authors are strongly advised to convey a more balanced discussion by referring to the book chapter by S O Duke: “Glyphosate: Uses Other Than in Glyphosate-Resistant Crops, Mode of Action, Degradation in Plants, and Effects on Non-target Plants and Agricultural Microbes”. Stephen O. Duke (2020). In: J. B. Knaak (ed.), Reviews of Environmental Contamination and Toxicology Volume 255, Reviews of Environmental Contamination and Toxicology 255, https://doi.org/10.1007/398_2020_53.
Response 2: We sought to adjust the discussion and conclusions so that there was no possibility of bias, and we considered the study cited above.
Point 3: SUGGESTION for future research on guarana: Authors are made aware of the real possibility that glyphosate residue may accumulate in the guarana product as a result of its use for weed control in the crop. Glyphosate residue in the product could result in rejection of product in the EU, for example, where strict MRL (maximum residue level) regulations exist -- REFER ABOVE BOOK CHAPTER BY S.O. DUKE.
Response 3: We include the possibility of more widespread damage to non-target plants from glyphosate use and point to future research to complete the management needs for this species. This information is in the introduction and conclusion topics.
Reviewer 3 Report
There is no indication of crop yield, so you can not assume that the low rates of glyphosate do not cause crop injury. Spraying glyphosate over plants of any type and trying to determine visually the extent of damage is not real research.
English is fair. Some editing would be appropriate.
Author Response
Dear reviewer,
here is our response to your comments:
Point 1: There is no indication of crop yield, so you can not assume that the low rates of glyphosate do not cause crop injury.
Response 1: We understand the reviewer's position, and perhaps the way it was exposed in the manuscript brought some confusion. Our conclusions suggest that the low dose did not alter the development of Guarana in either application during the evaluation period. But let's be clear, we can't extrapolate production.
Point 2 . Spraying glyphosate over plants of any type and trying to determine visually the extent of damage is not real research.
Response 2: The main results of the action of glyphosate drift are the symptoms of phytotoxicity, observed visually. Therefore, it was decided to collect these data and interfere in the phytosanitary aspect. It will be the first time that a scientific document of impact reports on the symptoms of the action of glyphosate in the culture of guarana and this serves as a basis for other species and communicates the need for improvements in the management of cutting. However, this will only be possible with the publication of these results in a journal with impact as molecules.
Reviewer 4 Report
Dear Authors,
See my point-by-point comments in the attached file.

Dear Editors,
Authors should revise the language scientific style using the correct terminology. Articles published in the journals of Weed Science Society of America will be very helpful.
Author Response
Dear reviewer,
here is our response for your comments:
Point 1: Line 17: Include scientific name in parenthenses when the species is first mentioned.
Line 24: A reference is missing. Additional comment: Press the space bar to separate words, this problem occurs in several lines throughout the whole manuscript
Line 26: weed management is necessary to make production viable.
Lines 26-29: No need to talk about non-chemical weed control options. Your study does not include any of them, it focuses only on herbicides and in particular, glyphosate.
Line 29: Which are the advantages (briefly)?
Line 35: In which crops? You should explain the situation in Americas where GMOs exist (glyphosate-tolerant crops) and in Europe where glyphosate is used exclusively for weed control in orchards and vineyards. In addition, there is no information on glyphosate. Site of Action, spectrum of action, history of use and current use in agriculture and especially in perennial woody crops should be somehow mentioned. The paragraph should have double length.
Line 46: use “However” instead of but. Too informal.
Merge lines 43-46 & 47-50 and edit the text in an appropriate way. State somewhere here in a clear way that “The objectives of the study were to evaluate the effects of different glyphosate doses on guarana biometric characteristics, phytotoxicity, and anthracnose severity.”.
Response 1: We took into account all of the adjustment points suggested by the reviewers and made changes to meet each adjustment point as they contributed to the robustness of the manuscript and the expression the study was trying to show.
Point 2: Lines 68-71: Provide us with details on seedling origin (company name), soil properties and chemical composition of leaves.
Line 272: to different doses of glyphosate
Line 273: RH: 74%
Line 274: Flow rate? Water amount?
Paragraph 4.2. Provide full info on the product (name/location of manufacturer). The same is noted for the sprayer and the nozzle used. Write it down with lowercase letters. Provide application rates expressed as g a.e. ha−1. Glyphosate is not expressed as g a.e. ha−1. Spray solution is missing. Application pressure is missing. Spray angle is missing. Paragraph 4.3. Write it down as Completely Randomized Design (CRD). Provide full details on the surfactant (see comments for 4.2). Do not forget the surfactant’s active ingredient and (subsequently) its application rate expressed as g a.i. ha−1.
Paragraph 4.4. How many samples were used for all these observations? Please use SI units throughout the whole manuscript (line 296). In addition, the title is “Guarana growth parameters”.
General comment: Merge all your assessments in a common paragraph named “Data Collection”.
Response 2: We seek to satisfy all suggestions for materials and methods as they contribute to replication research and as a source of methods for other cultures with Guarana characteristics.
Point 3. Define which data needed transformations? How did you check normal distribution? Shapiro-wil test maybe? Did you also perform homoscedacidity tests (Levene)?
In line 311: write it down as “at a significance level of a = 0.05”. In addition, did you repeat your experiment and include the factor of experimental runs in an intial ANOVA? If this ANOVA is available, you can sumbit it as supplemetary material.
Instead of performing regressions to detect glyphosate inhure on guarana, you should analyze your data as in dose-response experiments to estimate ED50/GR50 and ED90/GR90 (with slope/intercept/standard error values included as well). See two very important papers in the sector of Weed Science to get feedback: https://doi.org/10.1017/S0890037X00023253 AND https://doi.org/10.1614/WT-06-161.1
Response 3:
The reviewers cite fine tuning and insert statistical analysis information. A new ANOVA was performed on the data collected after the second application and the growth data was summarized, mentioning only the other variables because they were not significant and we understood that there was no need to show them.
We analyzed the data of the dose-response model and the data did not match, so we did a regression analysis presented in the manuscript.
Point 4. General comment for the whole manuscript: Revise language scientific style using the correct terminology. Articles published in the journals of Weed Science Society of America will be very helpful.
Response 4. We adjusted the scientific language according to the suggestions and improved the translation to make the manuscript clearer.
Round 2
Reviewer 1 Report
The opinion of this reviewer is that the manuscript has been significantly improved, so it can be accepted for publication in this journal
Reviewer 3 Report
Paper is improved with new additions.
Reviewer 4 Report
Dear authors,
the manuscript has improved. It is suitable for publication.